# Procurement practices and value for money in State Corporations in Kenya

**John Muturi Waci** [1]*, **Peter Wang'ombe Kariuki**[2], **Purity Mukiri Mwirigi**[1]

**1** Department of Business Studies, University of Embu, Embu, Kenya, **2** Department of Business and Economics, KCA University, Nairobi, Kenya

* johnmuturi30@gmail.com

**Data Availability Statement:** All relevant data are within the paper and its Supporting Information files.

## Abstract

Public procurement related expenditure is approximately fifty to seventy percent of the national budget of developing countries and accounts for almost a third of the gross domestic product. Cognizant of the significant funds committed in public procurement, the quest for value for money is critical. This study sought to determine the effect of procurement practices on value for money in State Corporations in Kenya. Specifically, the study investigated the effect of procurement planning, supplier sourcing, supplies management and E-procurement on value for money. Data collected from 87 State Corporations in Kenya was used in this study. The results of the study indicated that procurement planning, supplier sourcing, supplies management and e-procurement positively and significantly affect the value for money in state corporations in Kenya. The study concluded that proper procurement practices positively and significantly affect the value for money in state corporations in Kenya. The findings of this study contributes to literature by providing an empirical examination on the impact of procurement practices and value for money from a developing country perspective.

## 1. Introduction

Value for money is a highly desired result in any organization's procurement process since it is necessary to make the best use of the limited natural, economic, and social resources [1]. The main idea is to weigh costs and advantages in relation to the total cost of ownership. Because of its nature, public procurement entails all tiers of government making discretionary decisions [2]. A successful procurement system should help in attaining value for money, professionalism, accountability, transparency and fairness among other goals [3]. Effective public investment and economic growth are contingent upon the implementation of sound public procurement policies. Poorly managed public procurement procedures can waste public funds, impede development goals, increase costs, and transform public investments into significant political and economic liabilities [4].

The use of public funds by government organizations to purchase goods and services is known as public procurement (OECD, 2021). According to [2], procurement-related expenses account for between 50 and 70 percent of government budgets in developing nations.

**Funding:** The author(s) received no specific funding for this work.

**Competing interests:** The authors have declared that no competing interests exist.

Additionally, public procurement contributes up to 15% of GDP in affluent nations and nearly a third of GDP in impoverished nations. In many nations, public procurement accounts for even over 50% of total government spending [5]. Given the substantial financial commitments made in public procurement, the pursuit of value for money, or VFM, is essential. According to [6], VFM is the use of public funds in a way that guarantees the achievement of economy, efficiency, effectiveness, and equity.

To guarantee that the limited public funds are used wisely and that VFM is attained, numerous nations have made significant efforts and persistent modifications to fortify procurement units [1]. Adopting laws and regulations to direct public entities in their purchase of goods and services is one of the changes. The Federal Acquisition Regulation (FAR) was implemented in the United States of America (USA) and is utilized by all agencies for the procurement of goods and services. The Public Contracts Regulations, 2015 were implemented in Europe to provide guidelines for public procurement.

In Africa, the Public Procurement Act, 2003 was passed in Ghana; Tanzania passed the Tanzanian Public Procurement Policy of 2012; and Uganda passed the Public Procurement and Disposal of Public Assets Act, 2003. The Public Procurement and Disposals Act (PPADA), which was implemented in Kenya in 2005, was terminated in 2015. Establishing protocols for public entities' disposal and procurement was the aim of the PPADA. The act was designed to maximize economy and efficiency, encourage fair competition and ensure that rivals receive the same treatment, encourage integrity, justice, openness, and accountability, boost public confidence, and make it easier to support local business and economic development.

Malpractices continue to occur in public procurement despite the efforts of numerous nations to improve it. According to a report by [7] $390–400 billion is thought to be exchanged annually worldwide through illicit public procurement methods. An estimated $148 billion is spent annually on misconduct in Africa. About 70% of public contracts in Sub-Saharan Africa included malpractices, which drove up the cost of contracts by 20% to 30%. According to [8], procurement malpractices are projected to cause damages ranging from 10% to 25%, and in certain cases, up to 50% of the contract value. These large percentages and quantities highlight how urgent procurement changes are to stop malpractices.

Research on how procurement methods affect value for money has been spurred by the widespread instances of procurement malpractices. While [1] studied VFM in Botswana, [9] concentrated on VFM in Tanzania, [10] focused on public procurement governance in Uganda, [2] examined VFM in Ghana. The research, however, did not definitively identify the public bodies' procurement strategies or their efficacy in obtaining value for money. A number of other studies have also researched aspects of procurement procedures. For example, [11] studied procurement planning; [12] studied supply sourcing; [13] studied supply management. The studies, however, did not assess every procurement procedure, from pre-tendering to post-award. By examining the procurement strategies employed by State Corporations in Kenya and their impact on Value for Money, this study aimed to close these gaps and concentrated on procurement strategies from pre-tendering to post-award.

Specifically, the study sought to assess the effect of procurement planning on value for money in State Corporations in Kenya, determine the effect of supplier sourcing on value for money in State Corporations in Kenya, examine the effect of supplies management on value for money in State Corporations in Kenya and to evaluate the effect of E-procurement on value for money in State Corporations in Kenya.

The study makes several contributions to the literature on value for money and procurement methods. First, the study offers actual data on the connection between value for money and procurement procedures in state businesses in Kenya, a developing nation. Thus, from the perspective of developing countries, the findings shed light on the impact of procurement

procedures and value for money. Second, this study was conducted following the adoption of the Public Procurement and Disposal Act, which was amended in 2015, and the Public Procurement and Disposal Regulations, which were established in 2020 by the Kenyan government. In order to obtain value for money, public institutions should use the procurement techniques specified by the act. As far as we are aware, no research has been done to find out how these procurement procedures affect the value of money in Kenyan state enterprises. Thus, this study offers an empirical analysis of the impact of the different procurement processes together with recommendations that policymakers can use to evaluate and revise procurement regulations. Thirdly, the report offers managers and other stakeholders' suggestions on procurement strategies they may use to obtain value for money.

The remainder of this paper is organized as follows: Section 2 presents the study's background, Section 3 offers a theoretical review, and section 4 develops the study's hypotheses. Section 5 presents the research design, whereas Section 6 presents the empirical data and discussion, and Section 7 presents the summary and conclusion.

## 2. Background

The Kenyan government formed 167 state corporations to provide a range of services to its inhabitants [14]. The companies have been divided into groups based on the services they provide, which include training and research, public universities, financial, commercial, and manufacturing, service businesses, regional development, postsecondary education, and regulation. To fulfill their duty, the State Corporations are required to purchase goods and services.

The Public Procurement and Disposal Act (PPADA) of 2015 and the Public Procurement and Disposal Regulations of 2020, which provide that the entities must independently be responsible for all of their procurement decisions, serve as guidelines for the procurement functions of public entities. A procuring entity is required by the regulation to set up committees for tender opening, appraisal, negotiation, inspection, and disposal.

The procurement procedures and methods that must be followed are outlined in the [15]. Open tendering is the preferred form of procurement. Its procedures include requirement definition, procurement planning, source identification, vendor evaluation and selection, contract award, contract implementation, and payment for goods and services. Adopting e-procurement is mandatory for public entities in order to conduct procurement procedures. In order to automate and streamline government finance management procedures, the national government adopted the integrated financial management system (IFMIS) in 2003. Procure to Pay (P2P), a key element of IFMIS, aims to completely automate the procurement and payment processes while maintaining accountability and transparency as guided by The Kenyan National Treasury in the year 2014.

Even though laws governing procurement processes are in place, unethical behavior has undermined the progress made and resulted in the misappropriation of public monies. According to a public procurement regulatory authority (PPRA) report, there are notable variations and inconsistencies in the procurement procedures used by various public bodies. According to the PPRA audit report for 2017–18, 27 public entities had an average score of 55.5% and an average risk level of 44.5%. The findings suggested that some public entities might not be able to attain VFM since the compliance risk fell below the 60% cut-off level. (PPRA, 2018). Due to the firms' failure to attain VFM, donors have also expressed concerns and withdrew some support [16]. Reports from the Office of the Auditor General for a number of Corporations also highlight apparent problems, such as the purchase of inferior goods, projects that are on hold, delays in project execution, the purchase of goods or services at exorbitant prices, and spending that is not appropriately related to the goods or services purchased

[17]. Numerous instances of malpractice demonstrate how much money is wasted on the general public and how citizens are refused services. This begs the question of what procurement policies firms follow and how that impacts value for money.

## 3. Literature review and hypotheses development

This study adopted two theories and focused on examining four hypothesis as presented in this section.

### 3.1 Institutional theory

According to the theory put forth by [18], environmental factors have a greater impact on an organization's formal structure growth than do market forces. Institutional isomorphism is attained when an organization conforms its behaviors to the expected institutional environment. The institutional isomorphism process, according to [19], is a way for an organization to acquire legitimacy in response to three different kinds of institutional pressures: coercive, normative, and mimetic forces. Formal and informal forces that are used against the organizations by other strong organizations or entities that the organizations rely on apply coercive pressures. These pressures within the framework of procurement practices can take the shape of government rules and regulations requiring the use of particular procurement practices [20].

Professionalism and related networking create normative demands [19]. According to [21], organizations encounter normative pressures to be regarded as genuine by their peers in the professional community. According to [22], trading alliances, associations, and organizations' desire to be affiliated with them can thereby exert pressure on procurement practices. The Supplies Practitioners' Management Act of 2007 in Kenya assigns Kenya Institute of Supplies Management (KISM) the responsibility of ensuring professionalism through membership and licensing.

Organizations may experience normative constraints as a result of their social responsibility to their communities to appear morally upright [23]. Uncertainty causes mimetic pressures, which lead to firms trying to emulate other prosperous organizations [19]. Mimetic pressures are frequently caused by rivalry amongst firms over procurement procedures [20]. Public institutions in Kenya compete with one another for ISO certification and performance contracting ratings, which enhances efficiency and effectiveness because most organizations want to be evaluated favorably. [24] challenged the theory, arguing that it overemphasizes the explanation of institutions and institutional processes at the expense of the operational mechanisms of organizations.

Notwithstanding the critique, the theory emphasizes how the firm is impacted by outside forces and how businesses select and carry out plans and policies that are appropriate for their organizational domains [25]. Since the theory shows that an institution can be affected by its external environment to adopt particular practices, it is helpful in this study as it allows identification of the various procurement procedures that State Corporation is likely to embrace. Kenya's State Corporation procurement policies are governed by the 2010 Kenyan Constitution as well as laws such as the PPADA, 2015. The extent to which state corporations in Kenya have embraced procurement techniques, as well as the impact of these methods on the organizations' performance, remain unclear. Thus, this study looked into the state businesses in Kenya's procurement methods and how they affect value for money.

### 3.2 Resource based theory

According to the theory, which was put forth by [26], variations in the resource endowments of firms within the same industry might account for performance discrepancies. According to

[27], resources are assets that a company uses to carry out its plans. These assets might be intangible, such as software and organizational procedures, or tangible, such information communication equipment. In order to establish a competitive edge, managers of a company should incorporate and merge these resources into groupings that form competencies [28]. But according to [29], the theory has been critiqued for suggesting infinite regress and being applicable mainly to big businesses with substantial market power. Businesses are lured into an unending quest for ever-higher-order capabilities because they are counseled to pursue superior capabilities for innovation.

Though there is some abstract truism in this criticism, [29] pointed out that it does not contradict the RBV because it dissolves with the introduction of non-tangible resources, such as human resources. This theory is helpful to the research since it emphasizes how varied outcomes may be achieved by organizations depending on how well they use their resources, which includes developing efficient procedures. The theory discusses how an organization can accomplish variable financial management (VFM) by using procurement methods as one of the primary resources. This helps to explain how the independent factors affect the dependent variable. Thus, this study looked into how different State Corporations in Kenya handled their procurement, and how it affected value for money.

### 3.3 Procurement planning practices and value for money

Research has demonstrated how important procurement strategy is to getting value for the money. According to [6], the first task that establishes the framework for further procurement operations is procurement planning. An effective plan should outline the steps involved in hiring suppliers, including need assessment and identification, procurement method selection, acquiring required approvals, funding determination, and timeline scheduling. Whether State Corporations have incorporated these practices into their procurement plans will be examined in the current study.

An organization's adoption and use of procurement procedures are essential. The absence of an integrated planning process is a serious deficiency in the supply chain as it results in numerous inefficiencies like high safety stock, challenges in controlling seasonal demand patterns, inadequate demand forecasting, long planning horizons, and an inability to identify supply constraints related to capacity or material availability, according to a study by [11, 30]. Provides evidence to support these views in a research on how procurement planning affects institutional performance. A case study of Mombasa Law Court shown that performance is greatly impacted by cost estimation, need assessment, and quality criteria. According to [31] study, performance is greatly impacted by need identification, need specialization, budget, and cost estimations. The study examined the impact of procurement planning on procurement performance in Mombasa County, Kenya [32]. Discovered that the procurement planning techniques the Kenya National Police Service used had a significant impact on performance in another study on the impact of procurement procedures on the performance of corporate organizations in Kenya: a case study. Another study by [10] indicates that in Uganda, corruption in public procurement involves misapplication of the procurement procedures and deviation from standard legal frameworks in a variety of ways that include procurement planning distortions sustained by supplier collusion, shady cost computations by assessment teams, poor quality of products and services provided, and appalling performance of civil and construction activities. Thus, the current study will look into the procurement procedures used by Kenyan state corporations and how they affect value for money.

It is also clear that an organization's performance is not improved by having a good procurement plan if its requirements are not followed. There is a significant correlation between

procurement planning and procurement performance, according to a study conducted by [33] on the impact of procurement planning on the procurement performance of the Agricultural Development Corporation, Nairobi. According to the study's findings, procurement performance is positively impacted by having a procurement portfolio, effective logistics management, and adherence to procurement plans.

These conclusions are further supported by a study on the impact of procurement methods on the procurement performance of public sugar production companies in western Kenya by [34]. According to the survey, companies should improve their planning and make sure that the procurement strategies are followed. We hypothesize that:

$H_{01:}$ There is a positive relationship between procurement planning and value for money in state corporations in Kenya.

## 3.4 Supplier sourcing practices and value for money

The way that both the public and private sectors assess successful procurement is gradually changing. According to [35], judgments about economic efficiency have moved away from a price-only criterion of success and toward a multi-criteria approach that takes into account both price and a variety of quality-related factors. Both the financial and non-financial aspects of an offer should be taken into account when choosing who gets the contract. In public procurement, a value for money bid award is attained when a bid is given to a bidder who delivers the best price-quality ratio, as opposed to awards based on the lowest price or the lowest cost, as stated by [36], providing evidence in support of this viewpoint.

A study conducted by [37], concur that a supplier of goods or services shouldn't be chosen just because they offer it at the lowest price [38]. Contended that procurement decisions should be based on an ideal combination of quantity, quality, and price over the course of the project's lifecycle in order to obtain value for money, in addition to quality and price.

[12] Conducted a study on the impact of sourcing procedures on procurement performance in state companies in Kenya: a case study of the Kenya Bureau of Standards. The analysis confirmed that sourcing policies have a big impact on output. More specifically, performance is positively impacted by global, green, and multiple sourcing, and negatively impacted by single source. Supplier selection processes and procurement process management practices had a significant impact on the performance of the commercial state-owned enterprises under study, according to [39] study on the Effects of Procurement Practices on the Performance of Commercial Enterprises in Nairobi County. We hypothesize that:

$H_{02:}$ There is a positive relationship between supplier sourcing practices and value for money in State Corporations in Kenya.

## 3.5 Supplies management and value for money

According to a study by [13], a company needs to set up a management system, like an inspection and acceptance committee, to make sure that the products and services it purchases are delivered on time. According to the study, inadequate inspection of goods and services led to under- or non-delivery of some items or the delivery of subpar commodities. According to a study done in Tanzania by [9], contract management is also essential to achieving value for money, particularly while carrying out development projects. The study's main objective was to incorporate contract management techniques into Tanzanian public procurement in order to maximize value for money. The study highlighted the effectiveness of contract processes for quality control, time management, and cost control, which led to value for money.

Achieving value for money also requires careful monitoring and assessment of procurement policies and practices. The main obstacles to achieving value for money in public

procurement, according to the findings of a study by [2] on the measures of ensuring value for money in public procurement: A Case of Selected Polytechnics in Ghana, are insufficient procedures for monitoring and evaluating the procurement policy to ensure value for money. Accordingly, the study suggested that in order to guarantee value for money, management support for the value for money (VFM) program should be encouraged at all administrative levels, and procurement regulatory authorities should work with public entities to ensure compliance by closely monitoring and evaluating the procurement policy to ensure VFM. We hypothesis that:

$H_{03:}$ There is a positive relationship between supplies management practices and value for money in State Corporations in Kenya.

### 3.6 E-procurement and value for money

In order to promote transparency and fair competition, governments have been implementing e-procurement in an effort to limit the discretion of public authorities and civil servants that could result in the inefficient distribution of public funds and inadequate provision of public goods [40]. In a study by [40] examined how e-procurement was implemented in Denmark, the Netherlands, and Portugal as well as its effects on institutional quality. The results showed that whereas a similar reform in Portugal did not result in a higher control of corruption, the introduction of e-procurement is generally linked to a relatively stronger control of corruption in the Netherlands and Denmark.

According to research by [42], e-procurement improves company performance and strategic sourcing. According to [43], the mechanisms via which e-procurement improves supply chain performance can be represented by relationships, information sharing, and supply chain integration [44]. Looked into how e-government affected the amount of corruption in a wide panel of nations between 1995 and 2009 and discovered that e-procurement decreased corruption. We hypothesize that:

$H_{04:}$ There is a positive relationship between e-procurement and value for money in state corporations in Kenya.

## 4. Research design

### 4.1 Sample selection and data sources

A questionnaire was employed to obtain primary data for the study. Six sections made up the structure of the questionnaire. The respondents' background information was the main focus of the first section, and the data for each of the following four objectives was the main focus of the subsequent parts. In order to collect data on the current state of the phenomenon and provide a description of the variables in a given situation, this study used a descriptive research design, which involves asking participants about their beliefs, attitudes, behaviors, and values [45]. The 167 State Corporations in Kenya were the study's target population.

The head of each State Corporation's procurement department provided the information. Since the director of the procurement department is in charge of developing and implementing procurement procedures, they were selected to supply the pertinent data needed for the study. Additionally, according to the PPADA, 2015, public institutions' procurement operations must be overseen by a licensed procurement specialist. The State Corporations in Kenya are categorized into eight sectors as per Table 1. Hence, stratified random sampling was utilized since it improves the statistical efficiency of the sample and offers sufficient information for studying a population with different categories [45]. In order to determine an appropriate sample size based on the study's objectives, the formulae indicated in Eq 1 adopted from Slovin's method was used [46]. As a result, 118 companies made up the sample size for the study.

**Table 1. Sample size.**

| Sector | Population | Sample |
|---|---|---|
| Commercial & Manufacturing | 33 | 23 |
| Financial corporations | 17 | 12 |
| Training and Research | 14 | 10 |
| Service Corporations | 26 | 18 |
| Regional development | 6 | 4 |
| Tertiary Education & Training | 5 | 4 |
| Regulatory | 29 | 20 |
| Universities | 37 | 27 |
| **Total** | **167** | **118** |

Each sector was taken as a strata totaling to eight strata's and a representative sample from each sector or strata was randomly picked to makeup the total sample of 118 respondents as per Table 1.

$$n = \frac{N}{1 + N(e)^2} \tag{1}$$

Where **n** is the sample size, **N** is the total population and **e** is the error tolerance which is 5%

Thus n = 167/ [1 + 167(.05) 2] = 118

## 4.2 Research model and measurement of variables

The data for this study was gathered from the month of January 2023 to the month of May 2023. A research license was sought and granted by National Commission for Science, Technology and Innovation (NACOSTI) in Kenya to conduct the research. Written consent was also sought and obtained from the participants to provide the required data. The gathered data was thoroughly examined to make sure there were no inconsistencies and to look for any errors or omissions. To create a framework for analysis, the data was then classified into logical, descriptive, and significant categories. Both descriptive and inferential statistics were used to analyze the data. A correlation study was performed in order to ascertain the relationship between the variables. Next, a multiple linear regression model was applied to determine the association between value for money and procurement procedures. The study employed the regression model that is listed below.

$$y = \beta_0 + \beta_1 X_1 + \beta_2 X_2 + \beta_3 X_3 + \beta_4 X_4 + \varepsilon \tag{2}$$

**Where: y** was procurement practices, $\boldsymbol{\beta_0}$ was the Intercept constant, $\mathbf{X_1}$ is procurement planning practices, $\mathbf{X_2}$ was supplier sourcing practices, $\mathbf{X_3}$ was supplies management practices, $\mathbf{X_4}$ was adoption of e-procurement, $\boldsymbol{\beta_1} - \boldsymbol{\beta_4}$ was the corresponding Coefficients of independent variables and $\boldsymbol{\varepsilon}$ was the Error term

Table 2 presents an overview of how the variables were operationalized.

To assess the validity and dependability of research instruments, a pilot test was carried out. Ten percent of the sample utilized to collect data consisted of respondents who participated in the pilot test [45]. Thus, 16 firms, or 10% of the sampled population, participated in the pilot study. The state corporations were not among the entities that were looked at in the primary study; rather, they were chosen at random. To ascertain the consistency of the study instrument, a reliability test was conducted. The Cronbach Alpha Coefficient test, which suggests a

**Table 2. Operationalization of the variables.**

| Variable | Type of the variable | Indicators | Source |
|---|---|---|---|
| Procurement planning | independent | • Need Identification<br>• Cost Estimation<br>• Selection of procurement method | [6, 30] |
| Supplier sourcing | independent | • Invitation to Bid<br>• Bid Evaluation<br>• Award of Bid | [36, 37] |
| Supplies management | independent | • Inspection of Goods/Services/Works<br>• Storage and issues control<br>• Contract Monitoring, Evaluation and closure | [2, 9] |
| E-procurement | independent | • E-Tendering<br>• E-Ordering<br>• E-Payment | [41, 43] |
| Value for money | Dependent | • Economy<br>• Efficiency<br>• Effectiveness | [1, 2] |

threshold of higher than 0.7, was used to adopt the internal consistency technique [47]. Based on the test results, all the variables were kept for additional research because their Cronbach's alpha was more than 0.7. A validity test was conducted to ascertain the extent to which the data collection tool measures the intended variables [45]. Therefore, in order to verify the validity of the instruments, the opinions of specialists in the field were sought. This made it easier to revise and modify the study tools as needed to increase their validity.

# 5. Empirical results and discussion

## 5.1 Background information

The questionnaires were administered to 118 respondents who were in charge of procurement divisions at Kenya's State Corporations. A total of 87 completed and returned questionnaires were received. This resulted in a 74% response rate, which was deemed a representative sample for additional examination. In a related study, [48] suggested that a 70% return rate is excellent for analysis. Table 3 provides a background of the respondents. The results demonstrate that the respondents were informed because they had completed a variety of educational programs and, in the case of the majority (56%), held a degree. The majority of responders (58.6%), according to the statistics, had been employed for more than 15 years. As a result, the responses have expertise in their field and are able to supply the necessary information.

## 5.2 Correlation and diagnostic test results

The association between value for money and procurement methods in Kenyan State Corporations was investigated using Pearson correlational analysis. Table 4 presents the findings.

**Table 3. Background information.**

| Biographical Statistic | Biographical Variable | Frequency | Percentage |
|---|---|---|---|
| Education Level | Certificate/Diploma | 18 | 21% |
| | Undergraduate Degree | 49 | 56% |
| | Postgraduate Degree | 20 | 23% |
| Years worked in the Firm | 5 years or below | 8 | 9% |
| | 6 to 10 years | 12 | 14% |
| | 11–15 years | 16 | 18% |
| | over 15 years | 51 | 59% |

Table 4. Correlation matrix between the variables.

| | | VFM | PP | SS | SM | EP |
|---|---|---|---|---|---|---|
| **Value for Money(VFM)** | Pearson Correlation | 1 | | | | |
| | Sig. (2-tailed) | | | | | |
| | N | 87 | | | | |
| **Procurement Planning(PP)** | Pearson Correlation | .200** | 1 | | | |
| | Sig. (2-tailed) | .006 | | | | |
| | N | 87 | 87 | | | |
| **Supplies Sourcing(SS)** | Pearson Correlation | .323** | -.152 | 1 | | |
| | Sig. (2-tailed) | .002 | .161 | | | |
| | N | 87 | 87 | 87 | | |
| **Supplies Management (SM)** | Pearson Correlation | .235* | -.255* | .230* | 1 | |
| | Sig. (2-tailed) | .028 | .017 | .032 | | |
| | N | 87 | 87 | 87 | 87 | |
| **E-Procurement(EP)** | Pearson Correlation | .731** | -.306** | .151 | -.113 | 1 |
| | Sig. (2-tailed) | .000 | .004 | .163 | .299 | |
| | N | 87 | 87 | 87 | 87 | 87 |

**. Correlation is significant at the 0.01 level (2-tailed).

*. Correlation is significant at the 0.05 level (2-tailed).

The findings in Table 4 show that value for money and procurement planning had a positive and substantial correlation (r = .200, p-value<0.01). The findings also demonstrate a positive and substantial association (r = 0.323, p-value<0.01) between value for money and source of suppliers. The relationship between value for money and supply management also shows a positive association (r = 0.235, p-value<0.05). Furthermore, e-procurement and value for money had a positive and significant correlation (r = -0.731, p-value<0.01). Additionally, the correlation data show that there was no multi-collinearity because the correlation between the independent variables was less than 0.8 [49].

There are assumptions about the data utilized in linear regression analysis. Normality, the absence of multicollinearity, autocorrelation, homoscedasticity, and heteroscedasticity are the assumptions. The diagnostic test findings for linearity, multi-collinearity, autocorrelation, homoscedasticity, and heteroscedasticity are shown in this section.

A normal P-P plot was created to test for normalcy, and the results are displayed in Fig 1.

The residuals of a regression should have a normal distribution in regression models. The disparities between the observed and predicted values of the dependent variable are known as the residuals, or simply the error terms. The findings presented in Fig 1 suggest that the residuals in the normal Predicted Probability (P-P) plot follow the diagonal normality line, indicating a normal distribution of the residuals.

The absence of multi-collinearity between the independent variables is a prerequisite for regression analysis. To determine each variable's tolerance level and variance inflation factor, the independent variables were regressed. Table 8 presents the findings.

The tolerance and variance inflation factor (VIF) data for the variables are displayed in Table 5. According to the results, procurement planning procedures have a tolerance level of 0.847 and a VIF of 1.1180. The findings also indicate that the supply management tolerance level is 0.820 with a VIF of 1.219, the e-procurement tolerance level is 0.846 with a VIF of 1.183, and the sourcing practices tolerance level is 0.914 with a VIF of 1.094. Because the VIF values were below 10 and the tolerance threshold for each variable was over 0.1, it is clear that there is no multi-collinearity amongst the independent variables.

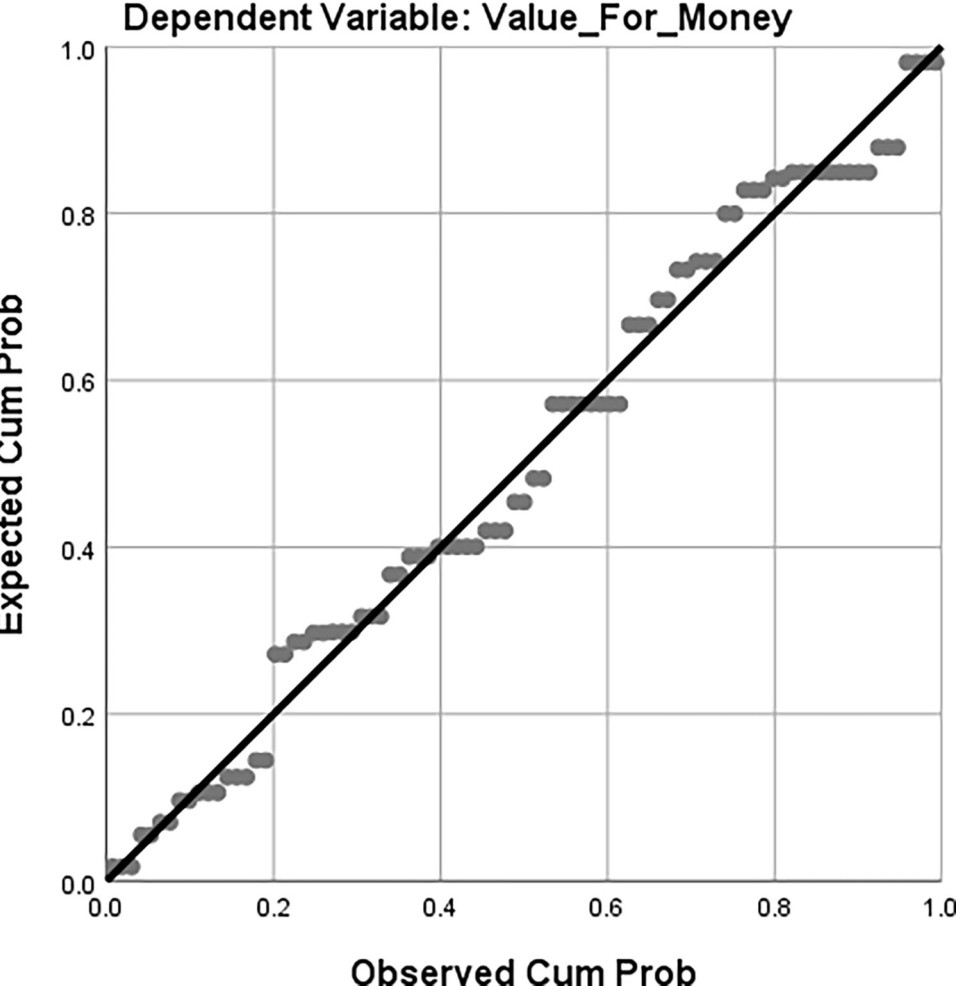

**Fig 1.**

Data for linear regression must have little to no autocorrelation. When the error terms or residuals are not independent of one another, autocorrelation arises. The null hypothesis, according to which the residuals are not linearly autocorrelated, was tested using the Durbin-Watson test. The Durbin-Watson statistic, represented by the symbol d, has a value range of 0 to 4, with 1.5 to 2.5 being the acceptable range [50]. Table 6 presents the findings.

**Table 5. Variance inflation factor (VIF) test results.**

| Variable | Collinearity Statistics | |
|---|---|---|
| | Tolerance | VIF |
| Procurement Planning | .847 | 1.180 |
| Supplier sourcing | .914 | 1.094 |
| Supplies Management | .820 | 1.219 |
| E-procurement | .846 | 1.183 |

**Table 6. Autocorrelation test results.**

| Model | R | R Square | Adjusted R Square | Std. Error of the Estimate | Durbin-Watson Statistic (d) |
|---|---|---|---|---|---|
| 1 | .821[a] | .674 | .658 | .52853 | 2.309 |

a. **Predictors: (Constant),** Procurement Planning, Supplier Sourcing, Supplies Management, E-Procurement.

b. **Dependent Variable:** Value for Money in State Corporations in Kenya

There is no first order linear autocorrelation, as indicated by Table 9's data, since the Durbin Watson statistic value (d = 2.309) falls between the two essential values of $1.5 < d < 2.5$. Heteroscedasticity, which happens when a variable's variability varies throughout a range of the other variable that predicts it, is a prerequisite for linear regression. Using the Breusch-Pagan test, heteroscedasticity was investigated. Heteroscedasticity was assumed to be absent in the null hypothesis and to be present in the alternative hypothesis. According to the decision rule, the null hypothesis is rejected if the significance level is less than 0.05. The test results are displayed in Table 7.

Table 10 presents the results, which indicate that the p-value for Breusch-Pagan is 0.602 and the p-value for Koenker is 0.489. The null hypothesis, which states that heteroscedasticity was absent, is accepted because the values for Breusch-Pagan and Koenker are both higher than 0.05. Therefore, the heteroscedasticity issue is not present in the study's variables.

## 5.3 Regression results and discussion

The study's hypotheses were tested using inferential analysis. Factor analysis was utilized in order to create variables for the regression data. The principle component analysis was specifically used to derive the regression constructs. The output of the main component analysis was tested for regression suitability using Bartlett's sphericity and Kaiser Meyer Olkin (KMO) sample adequacy tests. Since the KMO index was higher than 0.5 for every variable, the sample size was enough for additional study. The fact that every variable's p value for the Bartlett's test statistics was less than 0.05 further supported the appropriateness of the factor analysis.

The variables generated were procurement planning, supplier sourcing, supplies management and e-procurement. The regression results are presented in Tables 8–11 and Fig 2.

The results in Table 8 shows that the coefficient of determination ($R^2$) is 0.674 implying that the model estimated explains 67.4% of the variations in the value for money in state corporations in Kenya.

The analysis of variance (ANOVA) results presented in Table 9 demonstrate the significance of the association between value for money and procurement methods in Kenyan state

**Table 7. Breusch-Pagan and Koenker test results.**

|  | LM | Sig |
|---|---|---|
| Breusch-Pagan | 2.740 | 0.602 |
| Koenker | 3.427 | 0.489 |

**Table 8. Model summary.**

| Model | R | R Square | Adjusted R Square | Std. Error of the Estimate |
|---|---|---|---|---|
| 1 | .821[a] | .674 | .658 | .52853 |

a. Predictors: (Constant), Procurement Planning, Supplies Sourcing, Supplies Management, E-Procurement

**Table 9. ANOVA<sup>a</sup>.**

| Model | | Sum of Squares | Df | Mean Square | F | Sig. |
|---|---|---|---|---|---|---|
| 1 | Regression | 47.324 | 4 | 11.831 | 42.353 | .000<sup>b</sup> |
| | Residual | 22.906 | 82 | .279 | | |
| | Total | 70.230 | 86 | | | |

a. Dependent Variable: Value for Money in State corporations in Kenya

b. Predictors: (Constant), Procurement Planning, Supplies Sourcing, Supplies Management, E-Procurement

**Table 10. Coefficients<sup>a</sup>.**

| Model | | Unstandardized Coefficients | | Standardized Coefficients | t | Sig. |
|---|---|---|---|---|---|---|
| | | B | Std. Error | Beta | | |
| 1 | (Constant) | 1.561 | .526 | | 2.967 | .004 |
| | Procurement Planning (PP) | .828 | .072 | .790 | 11.521 | .000 |
| | Supplies Sourcing(SS) | .274 | .057 | .327 | 4.778 | .000 |
| | Supplies Management (SM) | .219 | .095 | .151 | 2.292 | .024 |
| | E-Procurement (EP) | .202 | .095 | .148 | 2.124 | .037 |

a. Dependent Variable: Value for Money in State Corporations in Kenya

**Table 11. Regression weights.**

| | | | Estimate | S.E. | C.R. | P |
|---|---|---|---|---|---|---|
| VFM | <— | PP | .828 | .070 | 11.798 | *** |
| VFM | <— | SS | .274 | .056 | 4.893 | *** |
| VFM | <— | SM | .219 | .093 | 2.348 | .019 |
| VFM | <— | EP | .202 | .093 | 2.175 | .030 |

**Where:** PP is Procurement Planning, SS is Supplies Sourcing, SM is Supplies Management, EP is E-Procurement.

businesses (F = 42.353, sig < .05). This suggests that the value for money in Kenya's state enterprises is greatly impacted by the procurement procedures. Therefore, it can be concluded from statistical evidence that Procurement Planning, Sourcing, Management, and E-Procurement are effective in forecasting the value for money in Kenyan state businesses.

The results in Table 10 provide the coefficients of the variables used in the study. The data was also analyzed using Structural Equation Model. The path analysis was generated and presented in Fig 2.

The regression equation model in this study is as shown in Eq 3.

$$Y = 1.561 + .828\ PP + .274\ SS + .219\ SM + .202\ EP \qquad (3)$$

According to the results, the constant term is 1.561, which means that 1.561 units of value for money may be obtained by setting all of the variables under examination to zero. This can be the result of additional variables that this study did not take into account. Procurement planning's regression coefficient is (.828, p < .05. This suggests that an increase of 1 unit in the procurement planning variable leads to an increase in value for money of 0.828 units when holding other independent variables to zero. This indicates that procurement strategy positively influences state companies in Kenya's value for money in a significant way. The

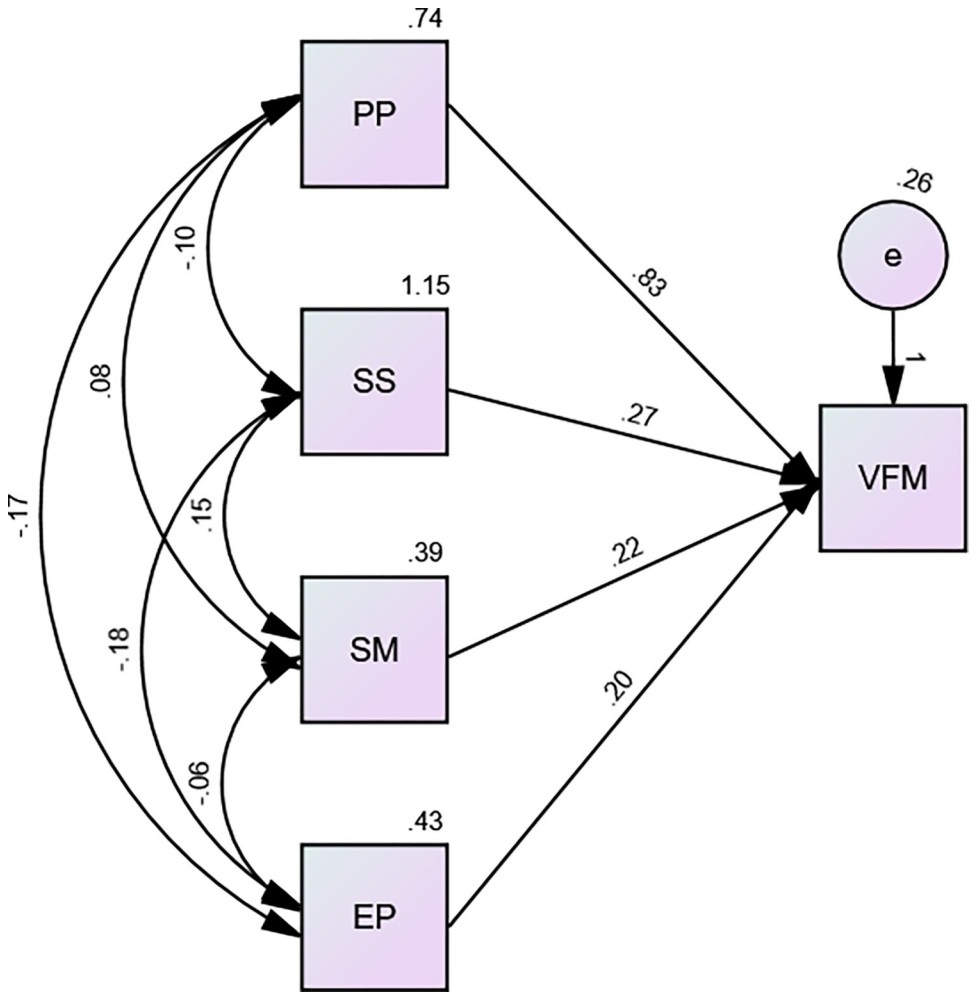

**Fig 2.**

outcomes are consistent with earlier research by [6] which showed that the main task that prepares the ground for later procurement operations is procurement planning. According to [30], the absence of an integrated planning process is a serious flaw in the supply chain because it results in a number of inefficiencies, including high safety stock, challenges in controlling seasonal demand patterns, inadequate demand forecasting, extended planning horizons, and an inability to identify supply constraints related to capacity or material availability. [11, 31–34] all corroborated these findings. The null hypothesis that procurement planning has no significant effect on value for money in State Corporations in Kenya was thus rejected.

The supply sourcing coefficient is (.274, p < .05). This suggests that when all other independent variables are set to zero, a one-unit increase in the supply sourcing variable will result in a 0.274-unit increase in value for money. The findings imply that supply sourcing procedures have a favorable and substantial impact on state firms in Kenya's value for money. The results are consistent with a study conducted by [12], which confirmed that sourcing procedures have a major impact on performance [39]. Confirmed in another study that procurement process management techniques and supplier selection protocols had a significant effect on a company's performance. According to [36], when a bid is given to a bidder who delivers the best price-quality ratio, rather than awards based on the lowest price or the lowest cost, a value for

money bid award is accomplished in the field of public procurement. The null hypothesis that supplier sourcing has no significant effect on value for money in State Corporations in Kenya was thus rejected.

The coefficient for supplies management is (.219, p < .05. This shows that a one unit increase in the supply management variable results in a 0.219 unit gain in value for money when all other independent variables are held at zero. It suggests that supply management techniques have a positive impact on value for money in Kenya's state corporations. The results are consistent with a research conducted by [9], which found that contract processes for quality control, cost containment, and time management were efficient and produced value for the money. According to [2], attaining value for money also requires careful observation and assessment of procurement policies and practices. The null hypothesis that supplies management has no significant effect on value for money in State Corporations in Kenya was thus rejected.

The coefficient for e-procurement is (.202, p < .05). This shows that a one unit increase in the e-procurement variable results in a 0.202 unit gain in value for money when all other independent variables are held at zero. According to the findings, e-procurement has a beneficial impact on State Corporations in Kenya's value for money. These findings align with research conducted by [43] as well as [42], which concluded that e-procurement improves company performance and strategic sourcing. According to a different study by [40], governments have been implementing e-procurement in an effort to maintain openness and fair competition by limiting the discretion of public servants and officials that could result in the inefficient use of public funds and subpar delivery of public goods. According to studies by [41, 44], e-procurement improved value for money and decreased corruption. The null hypothesis that adoption of E-procurement has no significant effect on value for money in State Corporations in Kenya is thus rejected.

## 6. Summary and conclusion

This study sought to determine the effect of procurement practices on value for money in State Corporations in Kenya. The specific objectives were to assess the effect of procurement planning on value for money in State Corporations in Kenya, to determine the effect of supplier sourcing on value for money State Corporations in Kenya, to examine the effect of supplies management on value for money in State Corporations in Kenya and to evaluate the effect of E-procurement on value for money in State Corporations in Kenya.

The regression results indicated that procurement planning, supplies sourcing practices and supplies management practices positively and significantly affect the value for money in state corporations in Kenya. The e-procurement positively influence the value for money in State Corporations in Kenya. The study concluded that proper procurement practices positively and significantly affect the value for money in state corporations in Kenya.

The study recommends that in order to ensure value for money, state corporations should prepare procurement plans and the same should be approved before the start of the respective financial year. The procurement plan should also involve all stakeholders. State corporations should also ensure compliance to procurement rules and regulations which require adherence to proper sourcing and supplies management. The corporations should also utilize e-procurement to undertake procurement process and enhance value for money. Government and policy makers should come up with policies that will ensure enforcement of procurement rules and regulations.

This study focused on the effect of procurement practices on value for money in State Corporations in Kenya. Future studies may focus on the influence of effect of procurement

practices on value for money in private or non- state corporations to find out whether similar findings can be achieved.

## Supporting information

**S1 File.**
(PDF)

## Author Contributions

**Conceptualization:** John Muturi Waci.

**Data curation:** John Muturi Waci, Purity Mukiri Mwirigi.

**Formal analysis:** John Muturi Waci.

**Funding acquisition:** John Muturi Waci.

**Investigation:** John Muturi Waci.

**Methodology:** John Muturi Waci.

**Project administration:** Peter Wang'ombe Kariuki.

**Resources:** John Muturi Waci.

**Supervision:** Peter Wang'ombe Kariuki, Purity Mukiri Mwirigi.

**Validation:** Peter Wang'ombe Kariuki, Purity Mukiri Mwirigi.

**Writing – original draft:** John Muturi Waci.

**Writing – review & editing:** Peter Wang'ombe Kariuki, Purity Mukiri Mwirigi.

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
