## [Decision Letter · Decision Letter 0]

9 Apr 2024

PONE-D-24-07847

Procurement Practices and Value for Money in State Corporations in Kenya

PLOS ONE

Dear Dr. WACI,

Thank you for submitting your manuscript to PLOS ONE. After careful consideration, we feel that it has merit but does not fully meet PLOS ONE’s publication criteria as it currently stands. Therefore, we invite you to submit a revised version of the manuscript that addresses the points raised during the review process.

Both reviewers acknowledged the paper's potential and provided valuable suggestions to further strengthen it.To ensure your research has the greatest impact, we kindly ask you to address the following key points raised by the reviewers:

**Focus on Sampling:** While mentioning the study population is informative, the reviewers request a more detailed explanation of the sampling procedures used to select the state corporations involved in your research. Describe the specific methods used to ensure a representative sample.**East African Focus:** Within the revised literature review, incorporate studies specifically from the East African region, including Kenya, Uganda, Tanzania, and others.**Recent References:** Update references throughout the literature review to ensure reliance on the most current sources.**Presentation Improvement:** Enhance the presentation of the methodology section by referring to top journals and articles in the fields of business and social science research.**Section Renaming/Merging:** Reviewer 2 suggests considering renaming or merging certain sections, particularly within the introduction, to enhance clarity and flow. Recheck references within the introduction for accuracy and consistency.**Abstract Revisions:** Move recommendations from the abstract to a separate paragraph within the conclusion section. Highlight the originality of your research within the abstract.**Comparative Analysis:** Compare your findings with results from previous similar studies in the domain of procurement practices and value for money.

We believe that addressing these points will significantly enhance the quality and impact of your research. We encourage you to revise your paper accordingly and resubmit it for further consideration.

We look forward to receiving your revised manuscript.

Kind regards,

Rateb J. Sweis

Academic Editor

PLOS ONE

Journal Requirements:

3. Please ensure that you include a title page within your main document. You should list all authors and all affiliations as per our author instructions and clearly indicate the corresponding author.

Reviewers' comments:

Reviewer's Responses to Questions

**Comments to the Author**

1. Is the manuscript technically sound, and do the data support the conclusions?

Reviewer #1: Partly

Reviewer #2: Yes

2. Has the statistical analysis been performed appropriately and rigorously? 

Reviewer #1: No

Reviewer #2: Yes

3. Have the authors made all data underlying the findings in their manuscript fully available?

Reviewer #1: No

Reviewer #2: Yes

4. Is the manuscript presented in an intelligible fashion and written in standard English?

Reviewer #1: No

Reviewer #2: Yes

5. Review Comments to the Author

Reviewer #1: Abstract

• Avoid mentioning the population of the study and instead focus on describing the sampling procedures used to select the involved organizations.

• Ensure that the sample size mentioned in the abstract matches the final size used in the analysis. Include only the data that was actually utilized, not what was initially planned.

Introduction

• Support the first sentence with relevant and current literature.

• Incorporate current literature throughout the section, not just in the first paragraph.

• Replace outdated sources, particularly concerning public procurement related expenditure, with more recent data from reputable sources such as the World Bank, OECD, or reputable journals.

• Clearly state the research question or objective that the study aims to address.

Background

• Utilize the information presented here to revise the introduction section, ensuring that the background section does not repeat.

Theoretical Literature Review

• Combine this section with the next one under the title “Literature Review and Hypotheses Development”.

• Include studies from the East African region, such as those from Kenya, Uganda, Tanzania, and others.

• Develop hypotheses that address existing gaps identified in previous studies, incorporating findings from the region for additional support.

Literature Review and Hypotheses Development

• Integrate this section with the previous one.

• Develop hypotheses that address gaps identified in previous studies, incorporating findings from the East African region.

• Update references to more recent sources.

Methodology

• Improve the presentation of this section by referring to top journals and articles in business and social science research.

• Ensure that Table 2 includes sources from existing literature.

• Consider using more robust measures of reliability and validity.

• Given the nature of collected variables, consider using PLS-SEM for analysis if there are latent and observed variables with the sample size.

Results and discussion

• Support findings with relevant literature rather than presenting numbers alone.

Conclusions, Implications and Limitations

• Ensure these sections are included and adequately presented.

Finally, make sure your document is proofread by a professional editor to reduce any typos and improve sentence structure.

Goodluck!

Reviewer #2: The topic of this paper is interesting, and with potential to dive deep inside, however, I have some suggestions as follows:

1) Some sections need to be renamed or/and merged and there is a need for recants references, especially within the introduction section.

2) The recommendations should be trimmed from the abstract and the originality should be included

3) The recommendations should be within a separate paragraph in the conclusion section

4) I recommend comparing the results with previous similar studies within this domain.

5)There are minor grammatical errors.

6. PLOS authors have the option to publish the peer review history of their article (what does this mean?). If published, this will include your full peer review and any attached files.

Reviewer #1: No

Reviewer #2: No

---

## [Author Response · Author response to Decision Letter 0]

26 Apr 2024

Dear Editor

This is in response to comments made by reviewers to the manuscript entitled “Procurement Practices and Value for Money in State Corporations in Kenya” by John Muturi Waci, Peter Wang’ombe Kariuki and Purity Mukiri Mwirigi. We have gone through the comments and made revision as per the attached Table.

Kindly note that we have updated the reference to include World Bank (2022), OECD, (2021); Lyra, et al.,(2022); Basheka, (2021). We have also removed from references OECD, 2015; OECD, 2017, World Bannk, 2012 and Lackert, 2009. This was to conform with reviewers comments.

We hope that you find our revised manuscript suitable for publication and look forward to hearing from you.

Sincerely,

John Muturi Waci

Business and economics, 

University of Embu

6-60100 Embu, Kenya

Email: johnmuturi30@gmail.com

---

## [Editor Report · Decision Letter 1]

2 May 2024

Procurement Practices and Value for Money in State Corporations in Kenya

PONE-D-24-07847R1

Dear Dr. WACI,

We’re pleased to inform you that your manuscript has been judged scientifically suitable for publication and will be formally accepted for publication once it meets all outstanding technical requirements.

Kind regards,

Rateb J. Sweis

Academic Editor

PLOS ONE

Additional Editor Comments (optional):

I am happy with the response quality by authors specifically to the first reviewer. I still think the manuscript can undergo another round of proofreading to improve the overall quality of communication. I wish the authors good luck in future work.
---

## [Editor Report · Acceptance letter]

10 May 2024

PONE-D-24-07847R1 

PLOS ONE

Dear Dr. WACI, 

I'm pleased to inform you that your manuscript has been deemed suitable for publication in PLOS ONE. Congratulations! Your manuscript is now being handed over to our production team.

Kind regards, 

on behalf of

prof. Rateb J. Sweis 

Academic Editor

PLOS ONE